# Continual familiarity decoding from recurrent connections in spiking networks

Viktoria Zemliak[1]*, Gordon Pipa[1,2], Pascal Nieters[1]

1 Institute of Cognitive Science, Osnabrück University, Osnabrück, Germany, 2 Frankfurt Institute of Advanced Studies, Frankfurt, Germany

* vzemliak@uos.de

## Abstract

Familiarity memory enables recognition of previously encountered inputs as familiar without recalling detailed stimuli information, which supports adaptive behavior across various timescales. We present a spiking neural network model with lateral connectivity shaped by unsupervised spike-timing-dependent plasticity (STDP) that encodes familiarity via local plasticity events. We show that familiarity can be decoded from network activity using both frequency (spike count) and temporal (spike synchrony) characteristics of spike trains. Temporal coding demonstrates enhanced performance under sparse input conditions, consistent with the principles of sparse coding observed in the brain. We also show how connectivity structure supports each decoding strategy, revealing different plasticity regimes. Our approach outperforms LSTM in temporal generalizability on the continual familiarity detection task, with input stimuli being naturally encoded in the recurrent connectivity without a separate training stage.

## Author summary

The ability to recognize a familiar stimulus without recalling its specific details is known as familiarity memory, and is fundamental to how animals and artificial agents learn and adapt to the environment. Our study explores how familiarity memory can be represented in a recurrent spiking network – a biologically-inspired computational model. Our model continuously encodes the familiarity of incoming stimuli in recurrent interneuronal connections via a local learning mechanism called Hebbian plasticity. We show that the structure of recurrent connectivity defines the firing activity of a network, and the familiarity information can be decoded from such activity using frequency (spike count) and temporal (spike synchrony) metrics. We demonstrate that a recurrent spiking network with unsupervised local learning recognizes familiar inputs across different timescales more accurately than an artificial neural network (ANN). Our findings advance the understanding of familiarity memory and suggest that spiking networks can

**Data availability statement:** All data in the work was programmatically generated. The code for generating data, running simulations and analyzing the simulation results, as well as the simulated data logs, plots and statistics, can be found at https://github.com/rainsummer613/spiking-continual-familiarity (https://doi.org/10.5281/zenodo.15547044).

**Funding:** The work was supported by funds of the research training group "Computational Cognition" (GRK2340 to GP) provided by the Deutsche Forschungsgemeinschaft (DFG, German Research Foundation) and the Volkswagen Foundation (96 572 to GP). The high performance computing cluster used to run the experiments was also funded by DFG (456666331 to GP). The funders had no role in study design, data collection and analysis, decision to publish, or preparation of the manuscript.

**Competing interests:** The authors have declared that no competing interests exist.

be useful in real-world tasks, such as continual learning in dynamic environments.

## Introduction

The brain can efficiently encode and later retrieve information about the stimuli encountered at various distances in time. Two processes contribute to the recognition memory: recollection and familiarity [1]. Recollection is oriented at the retrieval of detailed information about the stimulus from memory, whereas familiarity is a simpler process of detecting whether a stimulus has been encountered before. Dual process theory – the most prominent framework for recognition memory [2] – proposes that recollection and familiarity are functionally independent processes, mostly supported by distinct brain regions [3–5]. In the present work, we focus on modeling the familiarity component exclusively.

Familiarity plays an important and highly varied role in information processing across different timescales, ranging from retrieving childhood memories in lifelong learning to recognizing previously learned categories in continual learning to resolving the exploration-exploitation dilemma within individual learning tasks. Depending on stimulus familiarity, the brain may either encode novel stimuli as new memories or facilitate recollection of familiar stimuli [6–8]. Lifelong familiarity, or absolute familiarity, and familiarity on a shorter experiment timescale, relative familiarity, are associated with different patterns of neural activity. They are mostly observed in the anterior-temporal network which includes multiple regions, such as inferotemporal cortex (IT) and perirhinal cortex (PrC) [9–12]. Absolute and relative familiarity are associated with distinct event-related potentials: N400 and FN400 respectively [13] (but see also [14]), and different BOLD signal dynamics [15–16]. High absolute familiarity corresponds to an increased BOLD signal, whereas high relative familiarity leads to a decreased response – an opposite effect called repetition suppression. Repetition suppression was also shown on the earlier stage of visual processing in V2 [17]. A recent study in V1 revealed another pattern: an initial increase in the magnitude of visual evoked potentials by phase-reversed relatively familiar stimuli followed by their activity suppression [18]. The authors suggested that one mechanism responsible may be the increased inhibitory activity rather than long-term depression of excitatory synapses. Footprints of relative familiarity were also found in oscillatory activity: familiar stimuli are associated with low-frequency (10–15 Hz), and new – with high-frequency (up to 65 Hz) oscillations in V1 [19] and PrC [20].

Most computational models for familiarity recognition focus on relative familiarity, replicate repetition suppression as observed in PrC, and are based on feedforward [21–23] and convolutional [25–26] architectures with Hebbian or anti-Hebbian learning. In such networks, anti-Hebbian plasticity leads to higher memory capacity and naturally demonstrates repetition suppression effect [21,24–26]. In Tyulmankov et al. [24], a feedforward ANN with Hebbian and anti-Hebbian plasticity was first shown to outperform LSTM in the familiarity task of a continual nature. Read et al. [26] suggested that Hebbian and anti-Hebbian learning components may play complementary

roles in familiarity detection, better capturing relative and absolute familiarity respectively. Li et al. recently introduced an alternative implementation of Hebbian recognition memory with use of an energy-based approach in Hopfield Networks and Predictive Coding Networks [27].

Theoretical studies suggest that input familiarity can be encoded in lateral (recurrent) connections in spiking neural networks (SNNs) and detected by increased synchrony of network activity [28–29]. Therefore, we here investigate if familiarity can also be encoded effectively in a recurrent SNN, either in overall spiking activity (measured as spike count) or in synchronous spiking (measured as $R_{sync}$ ). We present three key contributions: First, we develop an SNN model for continual familiarity detection task that requires time-invariant familiarity detection. This is crucial, since familiar stimuli can be detected at various distances in time [24], albeit the detection accuracy may vary [26], and this time invariance has not yet been explored in spiking models. Second, we provide a systematic comparison of Hebbian versus anti-Hebbian learning continual spike-timing dependent plasticity (STDP) [30], demonstrating that, in a recurrent encoding of familiarity, Hebbian learning is a more robust mechanism. Third, we analyze the connectivity properties and plasticity parameters that emerge in successful Hebbian and anti-Hebbian recurrent networks, revealing that effective familiarity detection requires networks capable of clustering input patterns with well-separated strongly-interconnected clusters. We show that a synchrony-based readout for familiarity detection [28–29], as compared to a simpler spike count, is particularly sensitive to this clustering structure, and that Hebbian STDP achieves this network organization reliably.

## Results

### Encoding familiarity in lateral connectivity

The main mechanism for continuously encoding incoming stimuli in the lateral connections of our network is a continuous plasticity rule, which takes the shape of unidirectional, positive-only STDP (Hebbian learning) or unidirectional, negative-only anti-STDP (anti-Hebbian learning). Both plasticity rules are unsupervised and coincidence-based, relying on local spiking activity to update connections (Eqs 1–3). Our implementation uses an activity trace parameter, which determines updates based on the recent firing history of neurons.

$$\dot{A}_i = \delta \left(t - t_i\right) - A_i / \tau_a \tag{1}$$

$$\dot{w}_{ij}^+ = \delta \left(t - t_i\right) A_j \eta \tag{2}$$

$$\dot{w}_{ij}^- = \delta \left(t - t_i\right) A_j \eta w_{ij} \tag{3}$$

Eq 1 describes the dynamics of activity trace $A_i$ of neuron $i$, where $\delta$ is a Dirac delta function and $t_i$ is the spike time of neuron $i$. For this neuron, the activity trace thus gets updated on each spike ( $t - t_i = 0$ ) and decays with trace memory parameter $\tau_a$, which regulates how long information about a neuron's activity remains in the activity trace and can contribute to plasticity updates

Following Eq. 2, the Hebbian STDP update of each lateral connection after firing (post) of neuron $i$ is determined by the trace $A_j$ (pre) of each connected neuron multiplied with learning rate $\eta$ ( $\eta > 0$ for Hebbian learning). Our rule is unidirectional and positive-only (see Fig 1D), because input stimuli are encoded in firing rates, rather than spike order. It is therefore more important to emphasize the neurons which encode a single stimulus, and not their temporal firing pattern (see Continual familiarity experiments). Our rule strengthens co-activated neuronal ensembles representing common input patterns.

Eq 3 describes unidirectional, negative-only ( $\eta < 0$ ) anti-STDP: co-activation leads to weakening of synapses, thus promoting competition and decorrelation between neurons. Unlike positive STDP, anti-STDP weakens synapses and risks collapsing connectivity if not properly constrained. To prevent this, we implemented two regulatory mechanisms: i)

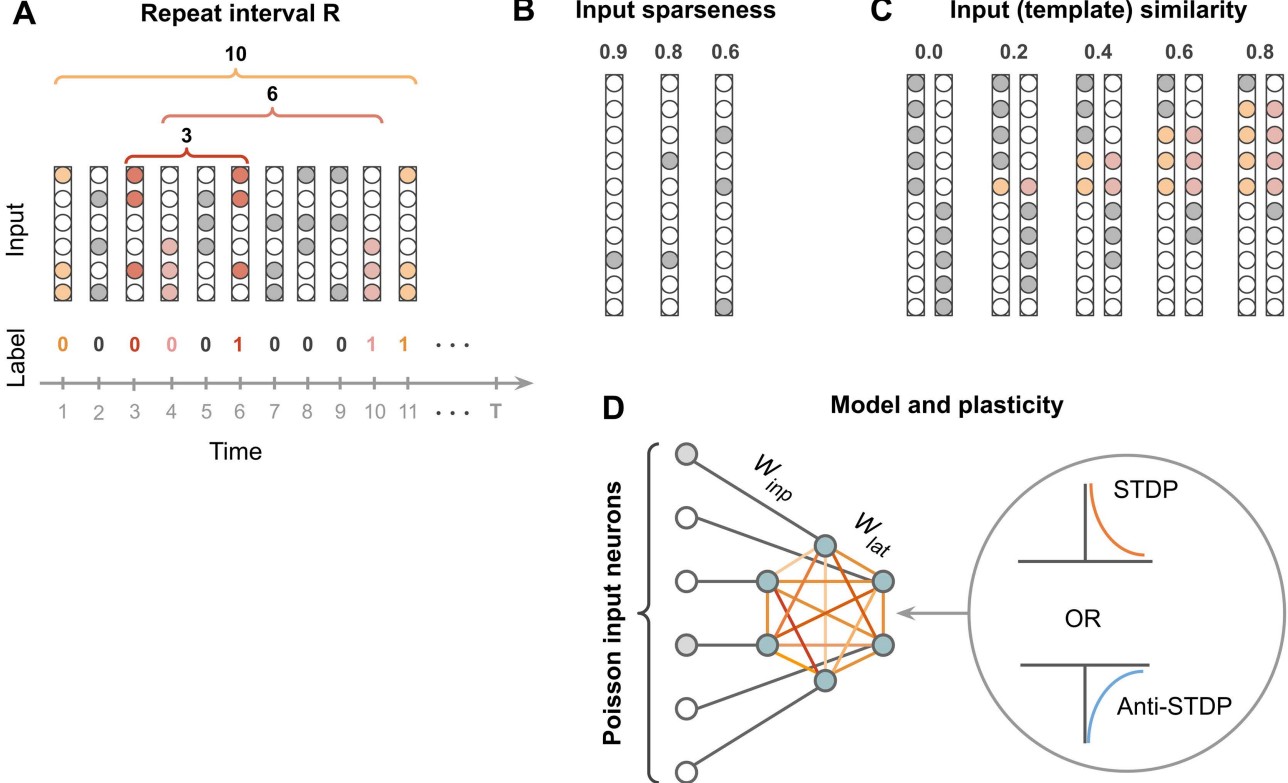

**Fig 1. Model and experiments.** A. Structure of the dataset: binary vectors are organized along a temporal scale. Some of the vectors repeat after *R* time steps, others are randomly generated. Repeated vectors are considered familiar and are labeled as 1, non-repeated vectors are non-familiar and labeled as 0. **B**. Three degrees of input sparseness are used in the experiment: 0.6, 0.8, 0.9. **C**. Inputs with correlation level modeled as template similarity, i.e., the fraction of externally stimulated neurons matching those in the same template activation pattern: 0.0 (main experiments), 0.2, 0.4, 0.6, 0.8 (see Methods Correlated inputs). **D**. The model architecture: each Izhikevich neuron [31] has a one-to-one connection to the spiking input. Izhikevich neurons are laterally connected to one another. Lateral connections undergo unidirectional STDP or anti-STDP.

a multiplicative rule that adapts weight proportionally and prevents abrupt weight erasure; ii) weight clipping to a minimal value, which prevents synapses from being driven to zero and preserves a functional baseline of recurrent connectivity.

For the network to remain expressive when performing a familiarity detection task, familiar and novel stimuli must evoke distinct firing patterns, which requires diversity in the lateral connectivity structure. However, under anti-STDP, synaptic weights tend to decay toward similar minimal values, erasing these distinctions and reducing the network's ability to represent differences in input. Conversely, under STDP, synaptic weights can grow uncontrollably, leading to runaway excitation and excessive, non-specific activity in response to any input. To preserve network expressivity and balance network activity, we implemented weight normalization as a homeostatic mechanism under both plasticity rules [32–33]. Weight regulatory mechanisms are of particular importance in a fully excitatory network model, since it does not have inhibition as an alternative way of stabilizing network activity. From the biological standpoint, weight normalization is motivated by the synaptic competition for limited resources and was shown to enable the emergence of functional connectivity ensembles when paired with positive Hebbian plasticity [34]. Our weight normalization rule is:

$$w_{ij} = \frac{w_{ij}}{\sum_{j \in J} w_{ij}} W_{total}$$

(4)

Here, $w_{ij}$ stands for a connection between neurons $i$ and $j$. $J$ is a subset of neurons which form incoming lateral connections to a neuron $i$. $W_{total}$ defines the sum of all incoming lateral connections of a single neuron. It controls the overall activity level in the network.

In the context of input representations within the lateral connectivity matrix, this homeostatic mechanism is aimed at scaling the continuous connectivity updates and embedding them into the internal representation space. It has an additional parameter: normalization interval, which regulates how often normalization events happen, and thus how quickly new memories are rewriting the old ones. Longer normalization interval allows for more plasticity updates to update the weights before they become rescaled, gradually erasing existing old memories.

**Continual familiarity experiments**

The network performance was evaluated on the continual familiarity task adapted from [24]. Every input stimulus is a 100-dimensional binary vector. The dataset is a sequence of 500 stimuli and generated as follows: each stimulus is either a copy of the stimulus present in the dataset $R$ steps ago, or a new randomly generated stimulus (Fig 1A). We refer to $R$ as a repeat interval: The number of new stimuli between two familiar stimuli in the dataset. Each stimulus can be new with a probability $p$, or familiar, i.e., a copy of a previous input, with a probability $1 - p$. For our experiments, $p$ was set to 0.5. Additionally, each stimulus is characterized by its sparseness, i.e., relative difference between the number of 0s and 1s in a binary 100-dimensional vector (see Methods Continual familiarity dataset). In our experiments, we used stimuli with sparseness 0.6, 0.8, and 0.9. Intuitively, sparseness is the opposite of vector denseness: fewer 1s make a vector less dense and hence more sparse.

Every element of the stimulus binary vector represents one external input to a single neuron in the model (see Fig 1D). The binary stimuli vectors are transformed into a spiking input via the Poisson point process with the firing rate 100 Hz for 1s in the vector, and 0 Hz (no spiking) for 0s in the vector. Thus, whenever the stimulus is encountered for the second time in the dataset, i.e., is familiar, it is again generated from the same binary vector via the Poisson process. Thus, familiar stimuli are represented by entirely different temporal patterns: only their rate component is preserved.

The task is as follows: a network is presented with one stimulus at a time and has to predict whether the stimulus is novel or familiar. Our model operates continuously over time, with permanently active, unsupervised plasticity. We simulate 1000 ms of firing activity and predict the stimulus familiarity from resulting spike trains. We empirically selected the detection timespan which provided a tradeoff between the accurate and fast classification. The performance in all experiments was measured as prediction accuracy balanced by a proportion of novel stimuli in the dataset (see Methods Measuring model performance).

We used a genetic algorithm to optimize STDP (activity trace memory, activity trace increase, STDP update scaling factor) and homeostatic plasticity (total incoming lateral weight, weight normalization interval) meta-parameters used in the experiments (see Methods Meta-parameter optimization). The parameters were optimized for different $R$ values (3, 5, 10) and input sparseness levels (0.6, 0.8, 0.9; see Methods Continual familiarity dataset), separately for frequency- and synchrony-based familiarity decoding methods. After optimization, we conducted a series of simulation experiments, to test model capabilities for continual familiarity detection on $R$ values from 1 to 30. This allowed us to evaluate model generalizability over time: whether it can detect familiar stimuli on various time distances in the past, even if optimized for a specific distance only. This generalizability was evaluated separately for spike synchrony and spike count decoding methods, and for different input sparseness levels.

For our experiments, we used a laterally connected excitatory spiking neural network with Izhikevich model dynamics [31] and either Hebbian or anti-Hebbian learning mechanism (LESH). Our network consists of a single hidden layer of 100 spiking neurons with all-to-all lateral connectivity (Fig 1D). Every neuron in the hidden layer follows Izhikevich dynamics, which describes how its membrane potential evolves over time (see Methods Izhikevich spiking model). When

the membrane potential exceeds a firing threshold, a spike event is registered. The membrane potential is influenced by incoming spikes, both from the external input and lateral connections from other Izhikevich neurons.

## Continual familiarity classification

In our study, we used two methods for classifying, or decoding stimulus familiarity from spiking network activity: spike count and spike synchrony. Spike count reflects overall network activity (see below) and can therefore measure repetition suppression or enhancement. We also investigated spike synchrony (measured as $R_{sync}$, see below) which was previously shown to encode familiarity and could reveal familiarity encoding different from a suppression/facilitation code. We use both metrics for simple threshold classification: we measure spike count or $R_{sync}$ in the model output spike trains and find a threshold for the prediction. The threshold for classification is determined through genetic optimization before final experiments (see Methods Meta-parameter optimization). For Hebbian (STDP) models, stimuli with the measurement value below the threshold are identified as novel, and stimuli with above-threshold values c as familiar. Conversely, in anti-Hebbian (anti-STDP) models stimuli with lower synchrony or spike count are classified as familiar, and higher – as new. For both metrics, we only considered a neural population receiving the external input. Our network was purely input driven, i.e., only those stimulated neurons were firing.

The first decoding method is the population spike count and does not depend on the temporal structure of output spike trains (Eq 5). The second metric we use is $R_{sync}$ (Eq 6). It estimates spike synchrony in a fully excitatory network, following [29–30].

$$SC(N, T) = \frac{1}{N} \sum_{i=1}^{N} SC_i(T)$$

(5)

$$R_{sync}(N, T) = \frac{\widehat{Var}(\langle A_i(t) \rangle_{i \in N})_{t \in T}}{\langle \widehat{Var}(A_i(t))_{t \in T} \rangle_{i \in N}}$$

(6)

Here, $N$ defines the externally stimulated neuron population, and $T = 1000$ ms is a time interval for measuring spike count or synchrony. Eq 5 shows calculation of a population spike count, measured as an average spike count of all neurons in the population.

In Eq. 6, $R_{sync}$ quantifies neural synchrony by comparing the variance of the population mean activity to the average variance of individual neurons. Each neuron's activation trace $A_i$ is computed by convolving its binary spike train with an exponential kernel $\exp(-t/\tau)$, using a time constant $\tau = 5$ ms This value lies within the commonly used 1–10 ms range for estimating spike synchrony [35], and is comparable to the timescale of an EPSP. The exact choice of a timescale has minimal impact on the final $R_{sync}$ result. Intuitively, if all neurons fire in perfect synchrony, their activation traces are identical, so the variance of the mean field equals the variance of each neuron, yielding $R_{sync} = 1$ If neurons fire independently, their individual variances remain high, but the mean field becomes flat due to averaging, resulting in $R_{sync} \to 0$. Unlike pairwise correlation measures, which are limited to comparisons between two neurons, $R_{sync}$ provides a scalable estimate of synchrony across neural populations of any size.

## LESH generalizes across repeat intervals

To evaluate the performance of the LESH model for familiarity detection, we compared it to the performance of LSTM [36] and HebbFF [24] networks (the infinite-data experiment). LSTM networks are typically used in machine learning for solving memory tasks. HebbFF is utilizing a similar mechanism for memorizing stimuli as LESH - fast Hebbian plasticity. It was also recently used for solving a continual familiarity task, and outperformed LSTMs [24]. We also compared familiarity detection in LESH models using either spike count or spike synchrony as the classification metric.

We genetically optimized LESH and trained HebbFF and LSTM networks for individual $R$ values 3, 6 and 10, and then evaluated their performance for a range of $R$ values from 1 to 30 (Fig 2). Both in the LESH and HebbFF model, weights were continuously updated in the experiments using a plasticity rule, while the LSTM network weights were fixed after training. Intuitively, LSTM networks use internal state to represent memory and learn to control this memory during training with fixed weight matrices. Continuous plasticity models (LESH/HebbFF) instead use the plastic weight matrix itself to encode memory. The LESH and LSTM models were evaluated on data of sparseness 0.8, and HebbFF on the data of sparseness 0.0. HebbFF encodes inputs as binary vectors of -1 and 1, which is not applicable to different sparseness values. LESH and HebbFF networks demonstrate better generalizability than LSTM, whereas the LSTM network's performance is superior on the $R$ value used for optimization.

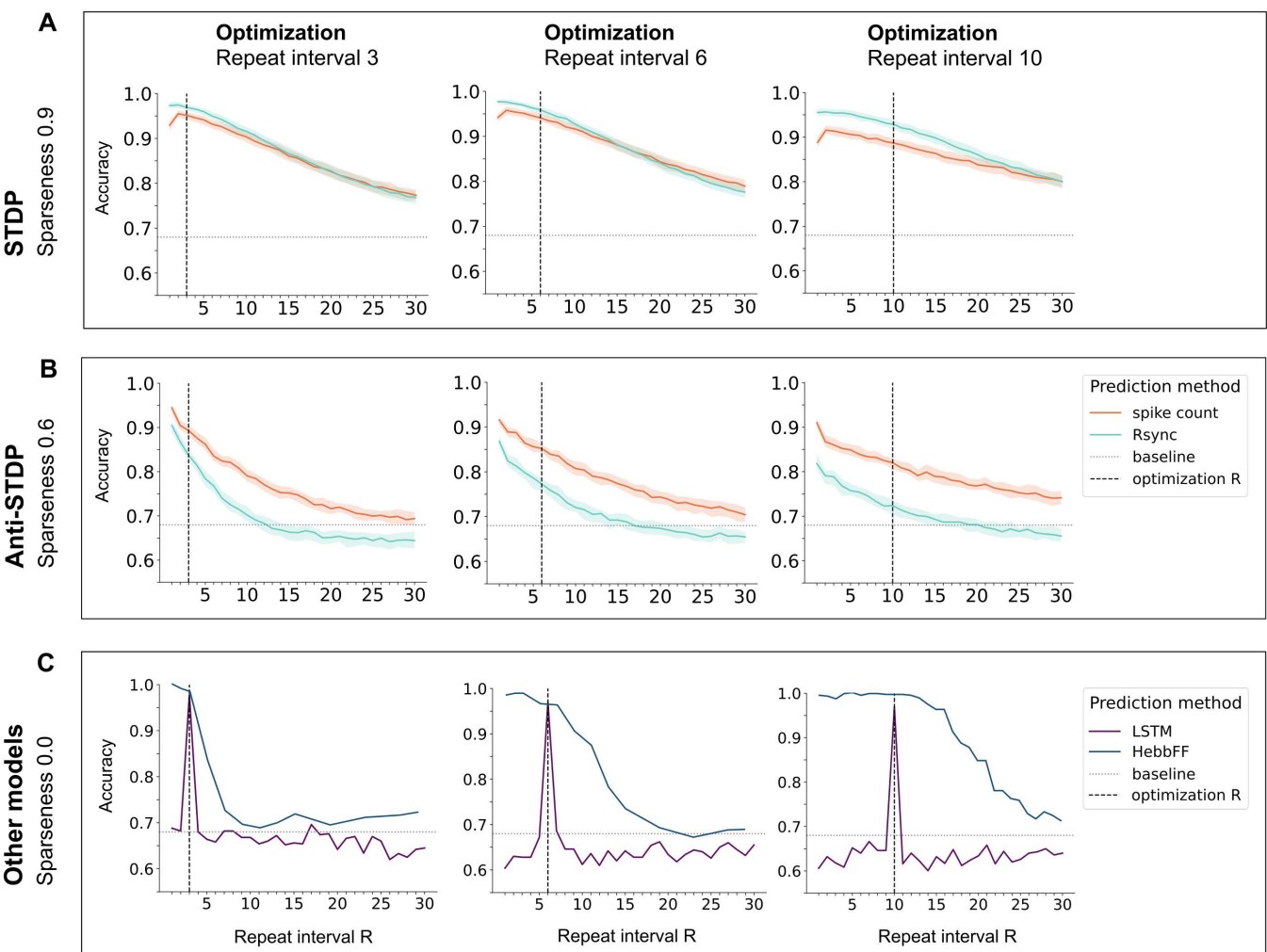

**Fig 2. Models generalization across repeat intervals on the continual familiarity task.** Performance of LESH with STDP and anti-STDP, HebbFF, and LSTM models across different repeat intervals RRR after parameter optimization. The baseline corresponds to predicting the most frequent class. For each model, parameters were optimized for a specific $R$, then evaluated across other $R$ values in the continual familiarity task. Results are shown for the best-performing input sparseness per model. **A:** LESH with STDP generalizes well, especially with sparse inputs. **B:** LESH with anti-STDP shows a similar generalization trend, though with lower overall performance. **C:** HebbFF (optimized in the infinite-data regime [24]) generalizes across $R$ values, while LSTM fails to generalize at all.

However, the LSTM network struggles to generalize across different repeat intervals because its internal memory representations might evolve arbitrarily over time. Memory representations in LSTM are stored in dynamic hidden states, which are not constrained mechanistically to maintain stable representations. Therefore, LSTM only learns to generate the appropriate memory representation at the specific time point which corresponds to the $R$ value used during training [24]. In contrast, the LESH model stores information directly in synaptic weights, which change in a consistent manner over time. Memory representations are constantly maintained in an accessible manner until completely rewritten by new overlapping memories, which enables familiarity detection at various repeat intervals.

For the LESH model, both spike count and spike synchrony classification strategies lead to a high generalizability across $R$ values. Spike synchrony has higher accuracy in the best case (highest sparseness) of Hebbian-based model, and spike count performs better under anti-Hebbian negative updates. Information about stimulus familiarity is encoded in both frequency and temporal characteristics of the spikes, but the spike count measure demonstrates higher accuracy in all of the conditions. In the best performing Hebbian-based models, STDP allows LESH to naturally remember stimuli encountered more recently and further in the past than the optimized target $R$ steps ago. For both spike count and synchrony, LESH performance gradually decreases with larger $R$ values, although the overall performance is noticeable higher for spike count. Thus, although model plasticity parameters were optimized for a specific repeat interval via the genetic algorithm (see Methods Meta-parameter optimization), the LESH network can flexibly access memories on various distances in time and shows state-of-the-art performance

When comparing LESH with HebbFF networks, the latter demonstrates either similar (for $R$ values 3 and 6) or better (for $R$) performance for the repeat interval it was optimized on. However, generalizability to other $R = 10$ values decreases more rapidly than the performance of Hebbian LESH. This is particularly striking for repeat interval 10: The HebbFF network demonstrates slightly better performance for $R < 15$, but then its ability to generalize declines steeply. Performance of the LESH network, however, declines smoothly and with a shallower gradient as $R$ increases, which shows superior generalization capability. We also compared the performance of the LESH network (performance of the LSTM network is identical across sparseness levels, therefore data is not presented). Sparser stimuli lead to better performance, but the ability to generalize over $R$ is maintained for all sparseness values (Fig 3) for Hebbian STDP. This is different in anti-Hebbian LESH models, which only show improvement over chance level in the least sparse conditions.

The LESH network's ability to generalize over repeat intervals highly depends on sparseness of the data, as well as on the repeat interval that its plasticity parameters were optimized for. The overall performance of Hebbian models indeed drops with decreasing sparseness of the input stimuli. This is because a certain number of internal representations needs to be stored in a network simultaneously, depending on the repeat interval. These representations overlap depending on the sparseness of the input. When many representations with large overlap must be stored in the network simultaneously, familiarity detection is more difficult. Consider, for example, a novel stimulus that already overlaps to a large degree with familiar stimuli already encoded in the lateral connectivity matrix. The overall activity in response to such a stimulus will be indistinguishable from the response to a familiar stimulus.

Anti-Hebbian LESH networks show the opposite pattern: dense input patterns are the only experimental condition leading to adequate performance, because sufficiently many connections between overlapping neurons become weakened more than non-overlapping connections. As a consequence, non-overlapping connections form the clusters able to influence the network activity, which is necessary for familiarity classification. On the contrary, anti-STDP leads to a distributed erosion of connectivity in sparse conditions, where nearly all synapses are similarly reduced. This is expected from the dynamical systems perspective, where recurrent networks with anti-Hebbian plasticity were shown to promote unstable activity due to non-consistent attractors which likely disrupts accurate classification [37] (see further explanation in Discussion).

Sensory representations of real-world stimuli are rarely random and, as a consequence, consistently different from each other. In addition to input sparseness and repeat interval in the dataset, we tested our models on the mutually

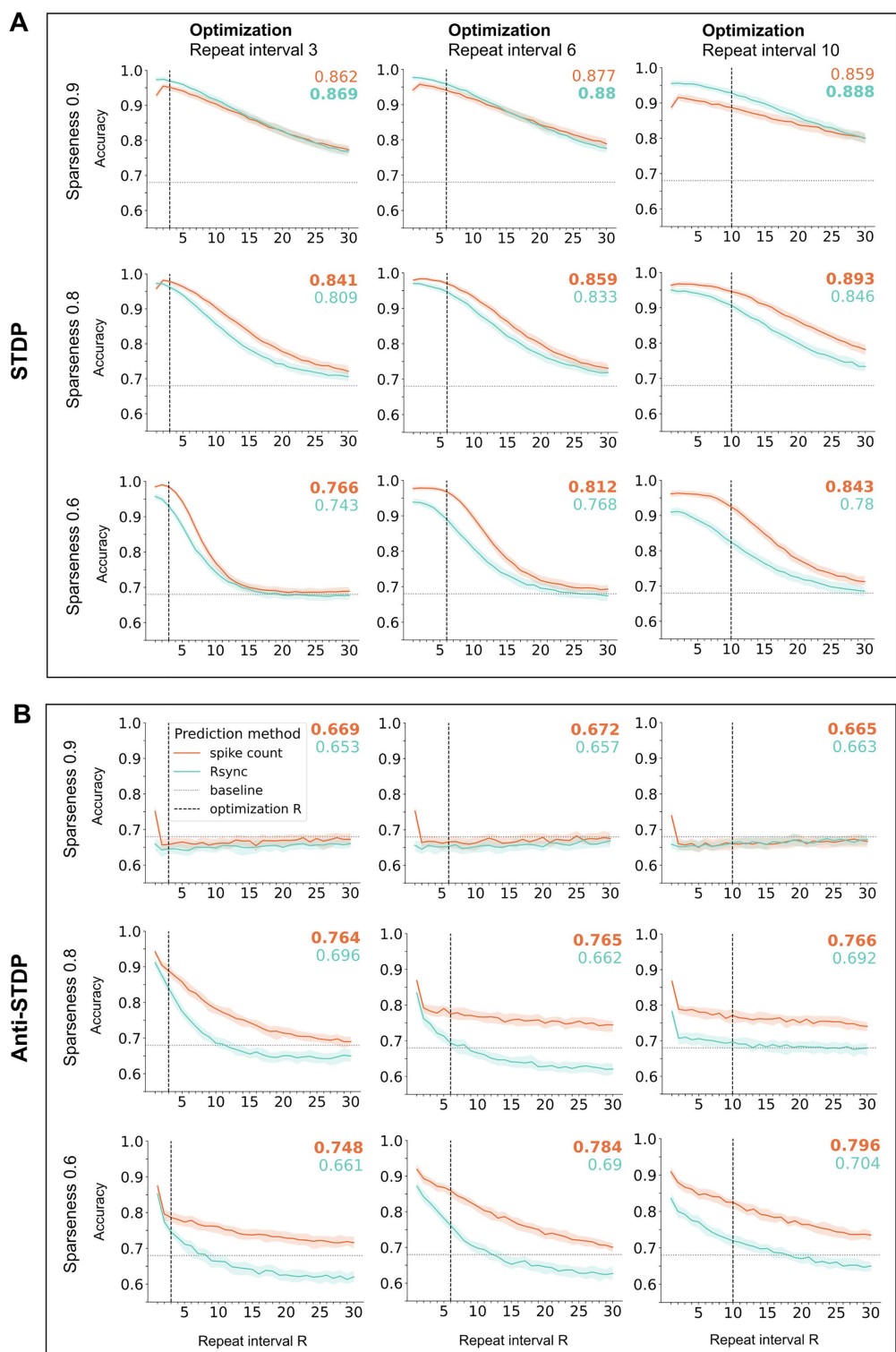

**Fig 3. LESH performance on data of various sparseness.** Colored numbers show generalizability: accuracy averaged across all test repeat intervals (larger number in **bold**). To ensure stability of the results, models were independently optimized for 20 times and yielded 20 parameter sets for every combination of sparseness and repeat interval values. The resulting plots were received after averaging accuracy for each combination. **A**. LESH with STDP performs better on sparse stimuli. **B**. LESH with anti-STDP performs better on dense stimuli.

correlated inputs, to see how they behave in more challenging input conditions (Fig 4, also see Methods Correlated inputs for generation of correlated patterns).

Hebbian LESH networks demonstrate the resilience to correlations in input data, retaining high generalizability even for moderately high similarity level 0.6, especially for sparse inputs. At the level 0.8, however, the performance drops to chance level. This appears to be a limitation on the representation space available for encoding incoming stimuli, which is reduced by high correlations (see Methods Correlated inputs). Hebbian learning benefits from as sparse and dissimilar inputs as possible, because these conditions lead to a formation of better separated clusters of interconnected neurons.

The performance of anti-Hebbian LESH models drops much faster: the network cannot perform familiarity classification for patterns with template similarity 0.6 and higher, and the performance for 0.4 is very limited. In contrast with Hebbian networks, anti-Hebbian models perform better under low sparseness. This interaction between inputs' sparseness and correlatedness might be worth investigating in future work.

## Plasticity regimes regulate representations learning

Our model's performance depends on its ability to distinguish new memories from representations already encoded in lateral connectivity, and hence on the ability of plasticity to encode these representations in a reliable manner for subsequent readout. We therefore analyzed network connectivity for a successful LESH network and how these network structures related to STDP meta-parameters. Optimal parameter combination allows for connectivity, which induces maximally different responses of a network to novel and familiar stimuli.

All plasticity meta-parameters in LESH were optimized using a genetic algorithm (see Methods Meta-parameter optimization) for a specific combination of repeat interval and input sparseness, thus they reflect how STDP adapts to different memorization needs. Plasticity is optimized to distinguish between overlapping memory representations for stimuli of low sparseness, and to keep more representation in memory simultaneously for longer repeat intervals. To analyze how plasticity shapes connections in a network, we computed Pearson correlations between its meta-parameter values and several characteristic features of the connectivity matrix, such as Gini index, transitivity, and betweenness-centrality [38]. These metrics describe clusteredness of the network, i.e., sizes, shapes and overlap of memory representations. We also compared how the plasticity direction (Hebbian or anti-Hebbian) interacts with the process of shaping connectivity (Fig 5).

**Gini Index** measures the inequality in synaptic weight distribution across the network. Under Hebbian plasticity, it is highest for sparse inputs and short repeat intervals (Fig 5A), where repeated co-activation strongly reinforces only a small subset of connections, creating unequal weight distributions. In networks shaped by anti-Hebbian plasticity, the trend is reversed (Fig 5B): Gini index increases for less sparse inputs because more co-activated synapses are repeatedly weakened, leaving only a few strong connections by chance. When repeat intervals are short, the same synapses are repeatedly co-activated and depressed, further accentuating this inequality. Thus, both plasticity types produce high Gini values via different mechanisms: one through focused connections reinforcement, the other through selective survival of few least overlapping connections.

Under Hebbian plasticity, Gini index strongly positively correlates with trace memory, learning rate and normalization interval, which promote starker plasticity updates for encoding new memories. Total incoming weight is the only negatively correlated parameter, since it allows more connections to grow big enough to sometimes allow connected neurons to fire even if they belong to different yet overlapping stimuli. For anti-Hebbian plasticity, Gini index retains strong positive correlation to trace memory, which regulates the size of updates and hence connection inequality, and is negatively correlated to minimal weight, which increases smallest connections thus equalizing the entire connectivity landscape. The Gini index only impacts the accuracy of Hebbian networks, with a stronger positive effect on the performance $R_{sync}$ than spike count, which suggests that $R_{sync}$ is more sensitive to independent representations.

**Transitivity**, which measures the prevalence of triangle motifs and reflects local clustering in the network, is only weakly affected by input sparseness in both Hebbian and anti-Hebbian networks. However, a slight increase in transitivity

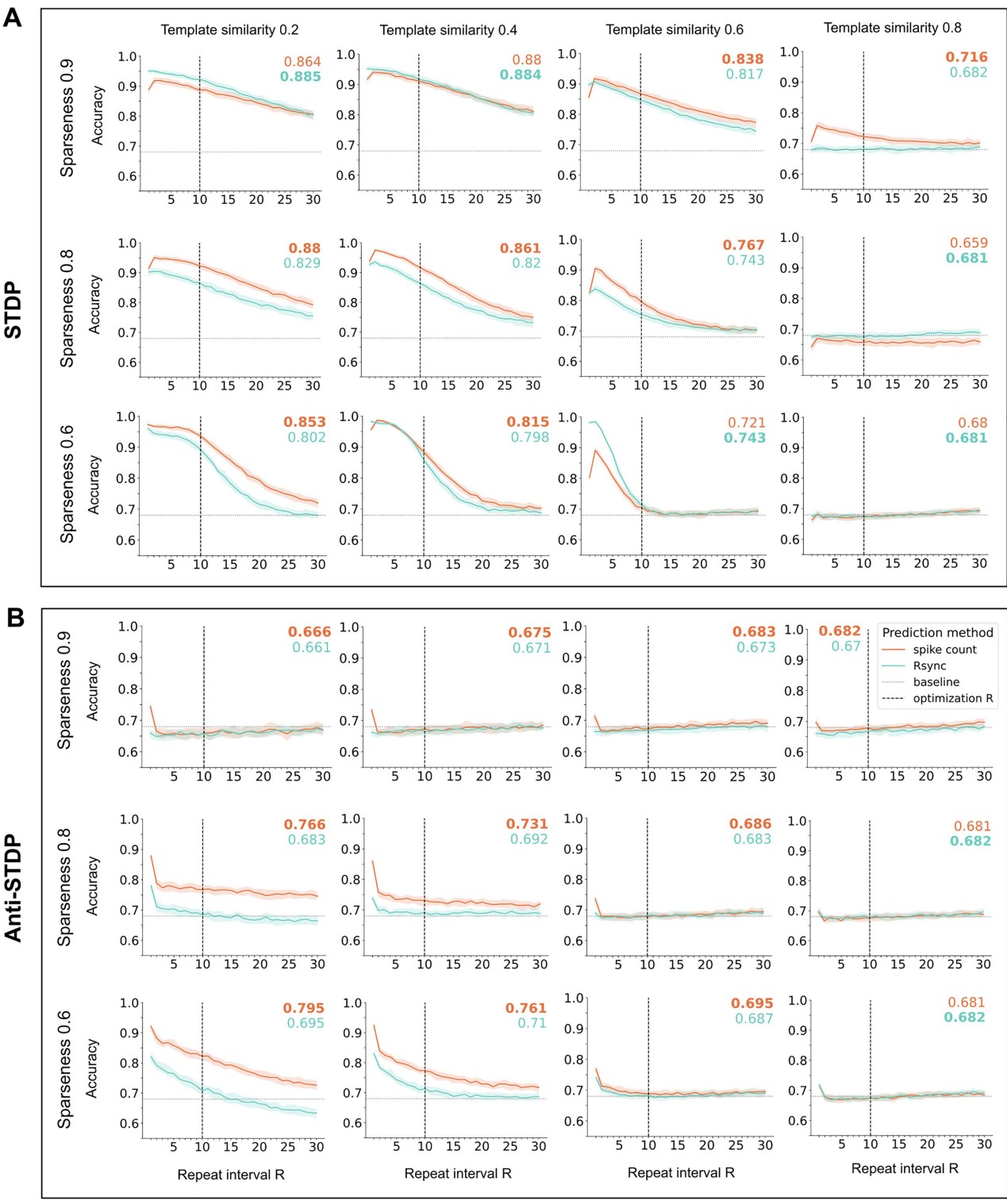

**Fig 4. LESH performance on correlated data.** To model input correlations, we generated input patterns with controlled levels of similarity to a pre-defined template (see Methods Correlated inputs). Colored numbers reveal generalizability: accuracy averaged across all test repeat intervals (a larger number in **bold**). To ensure stability of the results, models were independently optimized for 10 times and yielded 10 parameter sets for every combination of sparseness and repeat interval values. The resulting plots were received after averaging accuracy for each combination. **A**. LESH with STDP, optimized for repeat interval 10. These models are resilient to input similarities up to 0.6. **B**. The performance of LESH with anti-STDP, optimized for repeat interval 10 starts dropping already at similarity 0.4.

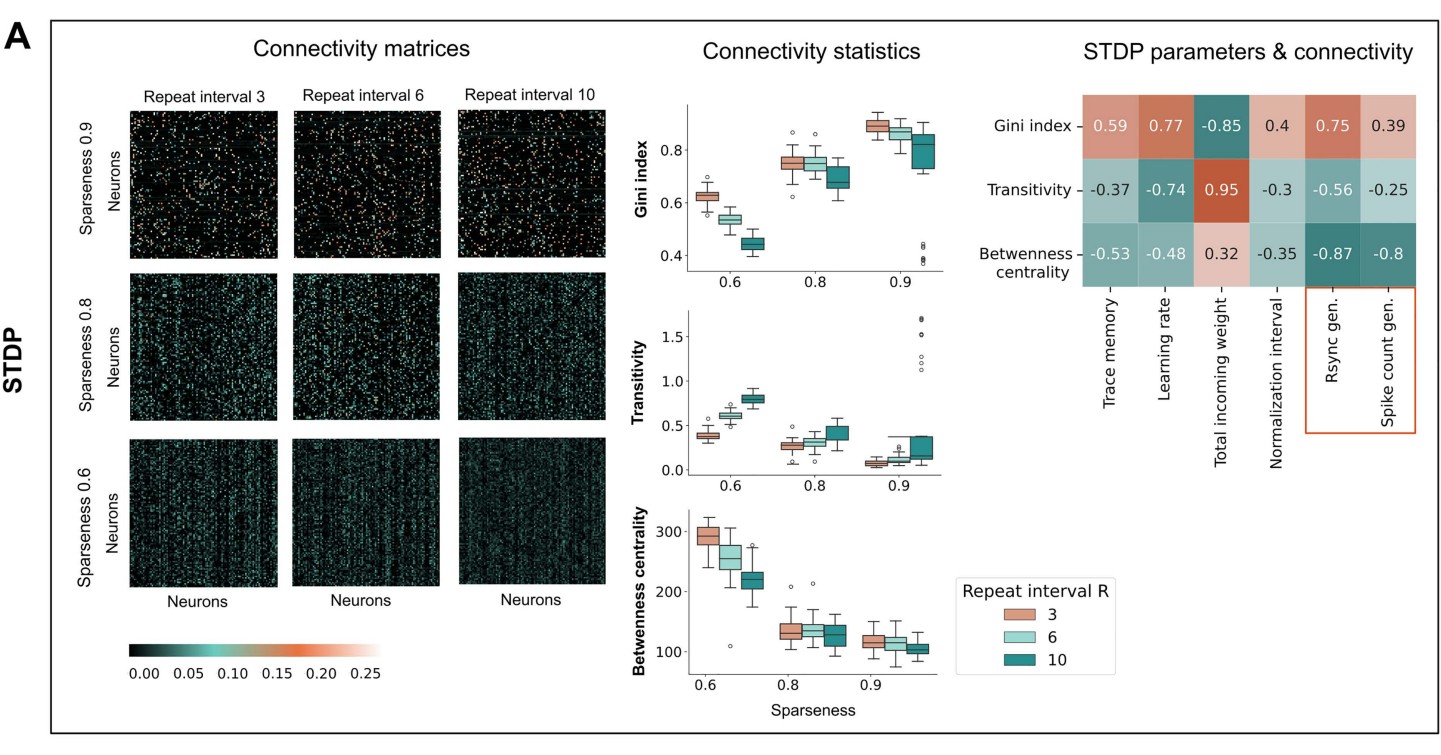

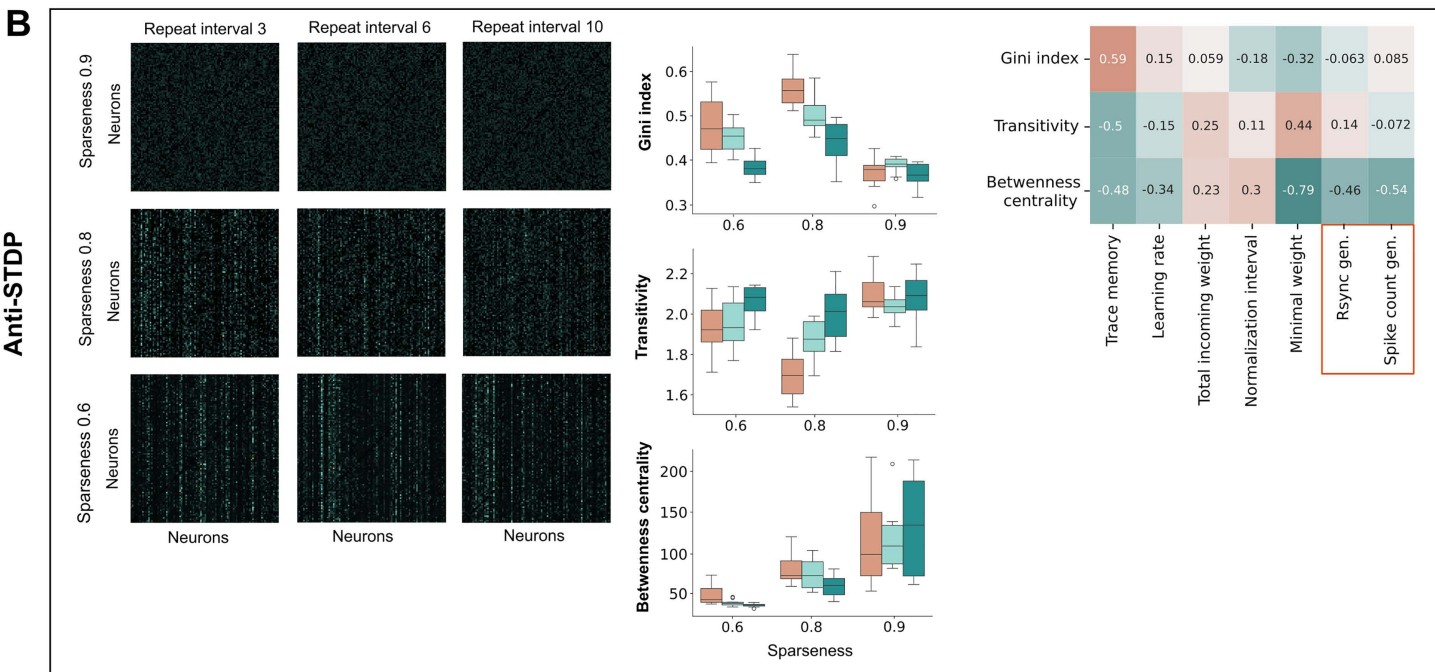

**Fig 5. Connectivity adaptation.** Various connectivity statistics of the optimized models. Data is presented for the LESH model optimized for spike count, after continuously encountering 500 input stimuli of various sparseness 20 times, for 20 independently optimized parameter sets, each corresponding to a repeat interval+input sparseness values combination. Note that statistics remain similar for the parameters optimized for $R_{sync}$. Weights were preliminarily rescaled via division by the sum of incoming lateral connections, optimized for each sparseness level and repeat intervals (see Eq 4). **From left to right**: connectivity matrices structure across input sparseness levels and repeat interval values; structural measures of connectivity; Pearson pairwise correlation between connectivity structure statistics and plasticity meta-parameter values + $R_{sync}$ and spike count generalizability values. **A**. For models with Hebbian plasticity, sparser inputs result in sparser connectivity, expressed in higher Gini index, lower transitivity and betweenness-centrality. All three connectivity features strongly correlate with the model performance, especially betweenness-centrality. **B**. Anti-Hebbian plasticity does not form clusters with distinctive Gini index and transitivity. Betweenness-centrality is the strongest predictor of model performance, reflecting lowest structuredness and hence worst accuracy for sparse inputs.

is observed with longer repeat intervals, because the updates between similar neurons encoding one familiar memory are less frequent – instead, they are interleaved by diverse updates for the other stimuli. In this case, plasticity is distributed across more pathways, rather than concentrated on the same ones, which may give triangle motifs a better chance to form incidentally.

In Hebbian networks, transitivity is reduced by all the same parameters that, in contrast, increase Gini index, since these parameters aim at better separating classes, and numerous local triangles dilute this separation. High total incoming weight is strongly correlated with transitivity since it promotes more neurons firing, including in response to false stimuli, which leads to the formation of synaptic connections between clusters. In anti-Hebbian networks, the length of memory traces negatively correlates with transitivity because longer traces induce more stark negative updates and thus rapid reduction of multiple connections. Transitivity moderately decreases the performance of Hebbian networks.

**Betweenness centrality** captures how often a neuron lies on the shortest paths between others, identifying key routes and bottlenecks in the network. In Hebbian networks, betweenness is highest with sparse input and short repeat intervals, where selective synaptic strengthening leads to the formation of hub-like neurons. Otherwise, connections become more evenly distributed. For anti-Hebbian plasticity, the trend is reversed: higher centrality emerges when inputs are more spaced in time, allowing a broader set of connections to persist and serve as hubs.

High betweenness-centrality signals about dense between-cluster connectivity through many weakly connected neurons between highly overlapping patterns, therefore it is also increased by the total incoming weight parameter and decreased by all other plasticity parameters which promote starker updates with Hebbian plasticity. In anti-Hebbian networks, it has notable strong negative correlation with minimal weight. Higher minimal weight values can flatten the network and reduce reliance on specific pathways, leading to more uniform routing and lower betweenness centrality, since no node is especially central in communication. In both network types, higher centrality corresponds to denser between-cluster connections.

Interestingly, under Hebbian plasticity $R_{sync}$ is more sensitive to changes in connectivity structure than spike count. It reflects fine changes in the network connectivity, implemented by STDP, and in the framework of our experiment these changes are inevitable since inputs are completely rate-encoded, without any temporal structure. These conditions are less challenging for spike count which does not rely on precise timing patterns. At the same time, in anti-Hebbian plasticity, co-active synapses are weakened, which disrupts synchrony by design: regardless of familiarity, often there is little synchrony to measure, and no meaningful network structure to amplify it.

Overall, successful familiarity detection by a recurrent network requires the formation of well-distinguished clusters with strong internal connectivity and weak between-cluster connections. This property is best reflected in low betweenness-centrality and a high Gini index. Hebbian networks naturally form such clusters while encoding strong connections between co-active within-pattern neurons. Since anti-Hebbian networks naturally destroy intra-pattern structures, they can only achieve this structure due to a sufficient amount of overlap, which decreases the connections between many overlapping neurons to such a degree that the remaining neurons become able to form clusters.

## Plasticity regimes depend on input sparseness

The optimal STDP regime – defined by meta-parameters that control aspects of the plasticity process – varies depending on the plasticity type, input characteristics and the classification strategy (Fig 6). Different levels of input sparseness lead to distinctly different optimal regimes, as sparse activity alters the number and timing of pre- and postsynaptic coincidences that drive the synaptic changes. The readout strategy (spike count or spike synchrony) also influences some optimal plasticity parameters, accounting for greater sensitivity of synchrony to recurrent connections. At the same time, different plasticity types require different interplay between all the parameters.

**In Hebbian networks**, all parameters which account for plasticity weight updates, namely learning rate, trace memory and normalization interval, are higher for the most sparse and thus least overlapping stimuli (Fig 6A).

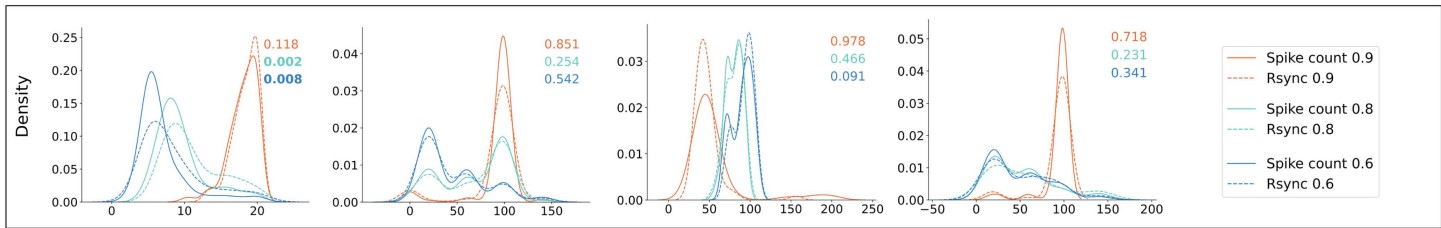

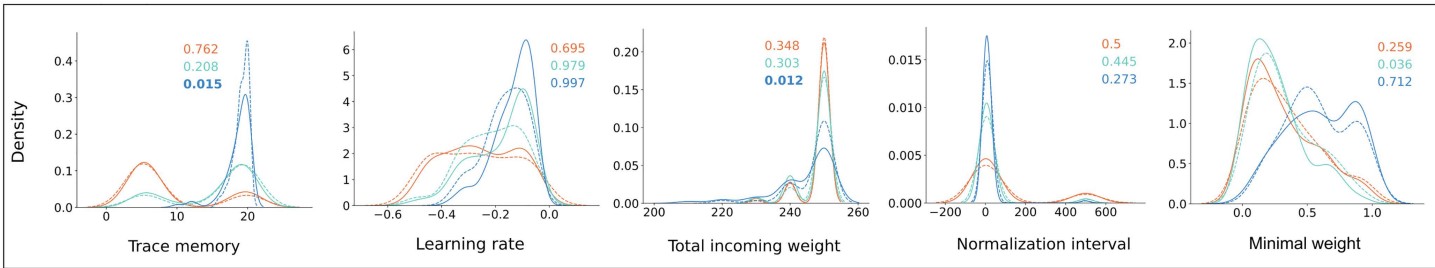

**Fig 6. Distributions of optimal parameter values across sparseness levels.** Parameter distributions more strongly differ across sparseness levels than familiarity detection measures ($R_{sync}$ and spike count). Each distribution, represented by a colored curve, includes 20 parameter values received during 20 independent optimization procedures. Color stands for sparseness: orange 0.9, green 0.8, blue 0.6. Colored numbers stand for p-values for differences between measures within every sparseness level, computed via a permutation test with 10000 permutations and Bonferroni correction for multiple comparisons. Data for plasticity types: **A**. Hebbian. **B**. Anti-Hebbian. Differences between $R_{sync}$ and spike count-optimal parameters are significant ($p < 0.05$) for trace memory in A and B, total incoming weight in B.

Greater updates naturally lead to better memorization and thus easier further familiarity detection: both spike count and spike synchrony threshold measures work most efficiently, when novel and familiar stimuli yield maximally different responses. We purposefully fixed the value of external input scaling factor (see S1 Table), thus the activity level in response to novel stimuli cannot be regulated much. During the optimization procedure, LESH had to identify the parameter sets which allow for maximal possible increased activity in response to familiar stimuli.

However, less sparse stimuli impose constraints on STDP weight update parameters, for two reasons. First, since less sparse input means more neurons receiving the external input encoding a single stimulus, the overall activity in the network has to be regulated. Second and most interestingly, when new and existing representations start to overlap for less sparse stimuli, novel and familiar stimuli might evoke falsely similar responses. Thus, weight updates have to be adjusted, which leads to decreased parameters regulating STDP updates, for less sparse stimuli. This markedly impacts spike synchrony, which requires stronger weights within a single representation to perform efficiently.

Trace memory is the parameter that differs most significantly between spike count and spike synchrony optimizations of Hebbian networks. It represents how long the activity traces of stimulus-encoding neurons remain strong *during the presentation of a single stimulus*. Longer memory traces allow neurons involved in a specific memory to form stronger connections with each other. This difference becomes significant for inputs with sparseness levels of 0.6 and 0.8, where the inputs overlap and, as a result, their internal representations in the network also overlap. Synchrony in a network depends on strong and even connectivity to quickly spread activation and make neurons fire coherently in time. When the overlap between internal representations increases at sparseness levels of 0.6 and 0.8, the $R_{sync}$ optimization uses longer trace memory to enhance synchrony within each familiar memory.

Total incoming weight is greater for less sparse stimuli because the number of neurons encoding each representation doubles at each sparseness level (see Methods: Continual Familiarity Dataset). For STDP, sufficient lateral input is needed to trigger firing, as memory representations form when neurons fire together and reinforce each other. Thus, the total input across all neurons in a representation increases.

**Anti-Hebbian networks** feature most widely distributed parameter values for the most sparse stimuli (sparseness 0.9), which reflects the failure of a network to find optimal parameter values leading to performance above chance level (see Fig 3B). For less sparse and thus less challenging anti-STDP conditions, optimal parameters feature notably shorter (up to every after every STDP update) normalization intervals in comparison to Hebbian-based parameter sets. This reflects the need to constantly boost overall net connectivity, so that the neurons would be able to influence each other's activity via recurrent dynamics, despite the rapid weight reduction. Total incoming weight is higher than in STDP-based optimal parameter sets, since there is no guided sparse strengthening of selected synapses and an alternative mechanism is required to keep some connections effective in terms of impacting the firing activity.

Trace memory and total incoming weight are the only parameters which are significantly different between sets optimized for spike count- and $R_{sync}$-based familiarity detection in anti-Hebbian networks. In both cases, these parameter distributions have less variance for $R_{sync}$-based detection for the least sparse stimuli (sparseness 0.6) − practically the only condition where anti-STDP leads to a somewhat reliable classification. As explained before, synchrony is more sensitive to changes in connectivity and struggles to perform under anti-STDP plasticity, which leaves it with a very small effective parameter window.

We also analyzed the optimal parameters for networks trained with different repeat intervals and input template similarity levels. In contrast to input sparseness, varying the repeat interval did not lead to statistically significant changes in the optimal plasticity parameters between $R_{sync}$ and spike count decoding strategies (see S1 Fig). Regarding input correlations (similarity), the only notable difference between the optimal parameters for the two decoding strategies was observed in the Hebbian models: optimizing for $R_{sync}$ resulted in a longer trace memory for highly correlated inputs, likely as a compensatory mechanism to enhance representation separation under high pattern overlap. No significant differences were found for the anti-Hebbian LESH models (see S2 Fig).

## Predicting LESH performance from connectivity features

Different plasticity regimes and input encodings lead to distinct patterns of network connectivity. This raises the question of whether the specific features of this connectivity predict how effectively the network responses distinguish familiar from unfamiliar stimuli. Therefore, we analyzed various connectivity features and their combinations to identify those with the greatest influence on LESH performance in a familiarity detection task. To achieve this, we trained and evaluated simple Decision Tree and Linear regression models using a 10-fold cross-validation procedure (see Methods Regression analysis for details). These models were then used to predict generalized performance of LESH in a continual familiarity task via spike count or synchrony, based on connectivity features (see Table 1). Note that we only conducted regression analysis

**Table 1. Predicting generalizability from connectivity.**

| Regression method | $R^2$ | RMSE | Gini index | Transitivity | Betweenness-centrality |
|---|---|---|---|---|---|
| Spike count | | | | | |
| Linear regression | 0.656 | 0.022 | 0.0 | 0.0 | -0.0004 |
| Decision tree | 0.67 | 0.019 | 0.136 | 0.092 | 0.772 |
| $R_{sync}$ | | | | | |
| Linear regression | 0.755 | 0.023 | 0.0 | 0.0 | -0.0005 |
| Decision tree | **0.845** | **0.017** | 0.149 | 0.033 | 0.818 |

for STDP-based predictions, since anti-STDP models often predict around chance level (Fig 3) and their performance is weakly correlated with connectivity parameters (Fig 4).

Predicting model performance generalizability in a continual familiarity task, using $R_{sync}$ and Spike count, from connectivity characteristics. Table includes $R^2$ and root mean squared error (RMSE) regression performance metrics and feature importances/coefficients for Decision tree and Linear regression methods respectively (see Methods Regression analysis for details on feature importances calculation). $R_{sync}$ performance can be predicted better from connectivity features, i.e., is more dependent on plasticity hyperparameters. Linear regression yields worse results than a non-linear Decision tree regressor, because predictor variables have complex non-linear interactions. Results in the table were received from 10-fold cross-validation, on 360 observations in total. See Methods Regression analysis for all hyperparameters and details on the regression procedure.

We used three characteristics of connectivity matrices for the regression analysis: Gini index, transitivity and betweenness-centrality, since all features were highly correlated with each other, and the addition of more than three features into the model had no impact on the performance. The particular set of predictor features was selected using grid search (see Methods Regression analysis). Both resulting performance metrics were substantially higher for the Decision Tree regression, suggesting that features have nonlinear interactions which cannot be captured by a linear regression model. Both models were able to better predict the $R_{sync}$ than spike count generalizability value from connectivity features.

Betweenness-centrality was the most important for prediction feature, for both spike count and $R_{sync}$ performances on a continual familiarity task. The predicted performance was negatively correlated with betweenness-centrality, which measures the representations overlap in a connectivity matrix. The Gini index played a bigger role in predicting $R_{sync}$ performance compared to spike count. This falls in line with our observation that trace memory, which is reflected in the Gini index and modularity of the connectivity matrix, is the only plasticity meta-parameter which differs significantly between networks optimized for spike count and spike synchrony read-outs.

The result that performance of the $R_{sync}$ network is very well predicted by network connectivity, and the result that prediction of the performance of the rate network is possible but significantly worse in comparison, highlights that spike synchrony is much more sensitive to features of network connectivity. This is in line with previous research showing how specific connectivity patterns lead to specific $R_{sync}$ responses [24] and may also help explain that robust performance was easier to achieve in rate networks.

## Discussion

We demonstrate that a recurrent spiking neural network with local unsupervised plasticity (LESH) successfully performs a continual familiarity task and generalizes well across repeat intervals. This result is consistent with previous research on continual familiarity in networks with unsupervised plasticity [24]. We show that a simple form of unsupervised Hebbian learning in a spiking network is naturally encoding input familiarity in lateral connectivity, without the explicit training and any top-down error or reward signal. The performance of our model depends on several factors: the direction of the plasticity rule (Hebbian or anti-Hebbian), the structure of the input (sparseness and correlation of the input representations), dataset structure (repeat interval used for plasticity meta-parameters optimization, and correlations of inputs in the dataset), readout method (spike count or synchrony). Although the LESH model demonstrates generalization capabilities across all aforementioned conditions, it performs best under Hebbian plasticity and sparse, decorrelated inputs.

The superior performance of Hebbian plasticity appears at odds with most familiarity models, which have shown greater memory capacity under anti-Hebbian learning [21,24–26]. However, these models are typically feedforward, where anti-Hebbian plasticity suppresses responses to repeated inputs by weakening strongly active connections – producing a familiarity signal without storing the input pattern itself. In contrast, recurrent networks rely on attractor dynamics, using synaptic weights to stabilize and reactivate internal representations. Anti-Hebbian plasticity disrupts these attractors, leading to unstable and less informative dynamics. This is consistent with recent work by Gong et al. [37], who showed

that while Hebbian learning in recurrent networks supports stable single-pattern equilibria, anti-Hebbian learning produces multiple equilibria and spontaneous limit cycles per pattern. Thus, our results suggest that the effectiveness of a plasticity rule depends on the network architecture: anti-Hebbian rules are beneficial in feedforward networks but can be detrimental in recurrent spiking circuits.

Hebbian LESH detects sparse (5–10% of neurons responding to each individual stimulus) less- to moderately correlated patterns more reliably, which is intuitive: lower overlap between stimulus representations results in more distinct recurrent connectivity patterns, reducing interference and improving the readout of familiarity. We do not see the model dependence on the input sparseness as a weakness, but rather as an inherent feature of a spiking dynamical system, which falls in line with biophysical observations of the brain. For example, only 0.5-3% of neurons in V1 of mice [39], ferrets [40] and monkeys [41] respond actively to individual natural images, which reveals sparse population coding.

We investigated two mechanisms for decoding stimulus familiarity from the model spiking activity: spike count and spike synchrony. Spike count is a simple frequency metric, whereas spike synchrony relies on temporal characteristics of the spike trains. Both metrics are viable for detecting stimulus familiarity, although spike count proves to be more precise and robust across all experimental conditions. However, synchrony outperforms spike count as a threshold classification measure for the most sparse inputs, which allow for almost non-overlapping representations. When the overlap is large, spike synchrony falls behind the spike count due to the inherent dependence on the strong and homogeneous representations. This property of synchrony falls in line with previous spike synchrony simulation studies [29–30]. The representations inevitably weaken to counteract the overlap, which disrupts synchrony. Spike count performance also decreases for the last sparse stimuli, but not as dramatically.

The influence of the plasticity algorithm, input sparseness and decoding method on the performance and network dynamics is reflected in the differences in the optimal plasticity parameters. For Hebbian (STDP) learning, the input sparseness is mediated by multiple parameters regulating the amplitude of plasticity updates, which are in turn balanced by the weight normalization across the neurons. The readout method shapes the connectivity organization: spike synchrony favors more homogeneous internal representations. On the contrary, anti-Hebbian models perform better with dense inputs, which allow the plasticity to boost several strong connections which least overlap among the inputs and start representing new patterns. Both plasticity types benefit from the use of longer repeat intervals during the optimization of meta-parameters, which get shaped to retain more representations between two repetitions as well separated as possible.

Neurons in our model are equipped with abstract yet strictly defined receptive fields: each neuron receives an input from a single corresponding spiking neuron. This is motivated by the need to study how the performance is influenced by recurrent connectivity patterns, excluding the confounding influence of overlapping input sources. Such simplified architecture does not reflect the complexity of real brain networks and does not aim to replicate any particular brain region, but nonetheless provides insight into how stimulus-selective neurons form memory ensembles shaped by gradually acquired experience, and how the system for time-invariant familiarity signaling emerges in their firing activity. Our best-performing model with positive Hebbian plasticity shows repetition enhancement: familiar stimuli elicit stronger network activity on re-presentation. This result contrasts with relative familiarity in PrC, where so-called novelty neurons exhibit repetition suppression, reproduced in a range of feedforward anti-Hebbian models [21–26]. Our model's behavior is more akin to either absolute familiarity in PrC, consistent with Read et al. [26], or relative and absolute familiarity in highly-recurrent areas engaged in pattern completion. For example, a range of studies identified groups of hippocampal neurons, traditionally associated with recollection, which fire more actively in response to stimuli of increased familiarity on various timescales [42–44]. V1 neurons also demonstrate initial repetition enhancement in a relative familiarity task [18]. In the said study, neurons' increased response to familiar stimuli is further suppressed during its prolonged presentation, which can be explained by the recruitment of inhibitory circuits. This subsequent suppression in LESH could potentially be reproduced in a LESH-like network with an addition of inhibitory circuits. Looking ahead, investigating the interaction between recollection and familiarity in rate and temporal domain is another prospective research avenue for recurrent excitatory networks.

Last but not least, our results suggest that spiking networks can be useful in continual learning tasks, since the input familiarity can be directly read out from their activity without an explicit training procedure, in a continuous fashion. For example, consider a classical continual learning paradigm where a model first must learn one task and afterwards is trained on another one. The main challenge for the model is to learn the new task without forgetting the old one. In this case, spike synchrony, or spike count, or another more complex metric could be used for detecting whether a stimulus has been encountered before and thus belongs to the old familiar task, or it corresponds to a new task. It has already been shown that spiking networks can excel in continual learning problems, due to the natural sparseness of their activity and a range of easily implementable methods to push this sparseness even further [45–46]. Moreover, it was recently directly demonstrated that task familiarity estimation in spiking neural networks can facilitate the efficient reuse of existing representations for new tasks, leading to improved performance and decreased energy consumption [47]. Therefore, we believe that frequential and temporal characteristics of spike trains can naturally encode the input familiarity, and this is a crucial feature for various kinds of continual learning.

## Materials and methods

### Continual familiarity dataset

A network is continuously presented with a set of stimuli. Every moment, one stimulus is presented, and this is considered to be a single dataset sample. A stimulus is a 100-dimensional binary vector consisting of 0s and 1s, and their proportion defines the sparseness parameter of the dataset. The more 0s and fewer 1s there are in a stimulus binary vector, the higher is the sparseness value (Eq. 7)

$$Sp = \frac{|d_0 - d_1|}{d_0 + d_1}$$

(7)

Here, $Sp$ is a sparseness parameter of the stimulus. $d_0$ and $d_0$ define the amount of 0s and 1s in a binary stimulus vector respectively. Note that $d_0 + d_1$ defines a dimensionality of the input vector, which also equals the amount of Izhikevich neurons in the model. In our experiments, we used three combinations of $d_0$ and $d_1$ values, corresponding to three levels of sparseness (see Table 2). A dataset consists of stimuli of the same sparseness.

Each stimulus is randomly generated with probability $1 - p$, and with probability $p$ it repeats the stimulus from the time step $t - R$, i.e. $R$ time steps ago in the dataset. We restrict every stimulus to appear no more than 2 times within a single dataset, hence a single stimulus can only repeat once. We generate our dataset with a new stimulus probability 0.5, thus the fraction of familiar stimuli is 1/3, and the fraction of novel stimuli 2/3 respectively. The procedure for the dataset generation was first presented in [24].

In our setup, a single dataset has a fixed repeat interval $R$, and the model performance is evaluated on multiple datasets with $R$ values from 1 to 30. Before the experimental simulations for performance evaluation, the model undergoes the optimization stage with a fixed $R$ value (see Methods Optimization for details). In our experiments, $R$ for optimization is fixed to either 3, 6, or 10. The evaluation is performed for multiple $R$ values, to estimate the model generalization ability.

**Table 2. Stimulus sparseness.** The proportion of 0s and 1s in the binary stimulus vector defines sparseness of the stimulus in a dataset.

| $d_0$ | $d_1$ | Sparseness (Sp) |
|---|---|---|
| 80 | 20 | 0.6 |
| 90 | 10 | 0.8 |
| 95 | 5 | 0.9 |

## Hyperparameter optimization

In our spiking network, the Izhikevich neuron model parameters are fixed, and several Hebbian plasticity and connectivity parameters are subject to the optimization, which takes place before the experimental simulations. The following parameters are optimized: lateral input scale, plasticity scale, total lateral input, weight normalization interval, and either spike count or $R_{sync}$ classification threshold. In the present study, optimization was performed with a variation of a genetic algorithm [48] (Fig 7).

A pipeline of the genetic algorithm. There are 12 parameter sets in every generation, i.e., in the iteration of the algorithm. Spike trains are generated with an Izhikevich model and with each parameter set in the generation. Then, accuracy of familiarity detection is measured with the use of spike count and $R_{sync}$ decoders. Those parameter sets which lead to the best performance, are selected for the next generation of the genetic algorithm. They undergo operations of crossover and mutations; also, new parameter sets are generated. The algorithm finishes if any of the termination criteria are met: 200 generations are over; accuracy 1.0 is achieved for any parameter set in a generation; accuracy has not increased for 15 generations.

The algorithm was run for 300 generations, each generation has 12 descendants (i.e., parameter sets), and for every generation the following operations were performed:

1) Each parameter set is used to evaluate the model on a dataset of $N = 300$.

2) The performance of the simulations with all parameter sets from the current generation is evaluated and compared. Note that we separately optimize the parameters for $R_{sync}$ and spike count metrics.

3) Mutation: in each of 3 best-performing parameter sets, one random parameter is multiplied by a random factor from 0.75 to 1.25.

4) Crossover: from 3 best-performing parameter sets, 3 pairs are formed. In each pair, a random half of parameter values are taken from one set, and the rest – from another.

5) Generation: 3 new random parameter sets are generated.

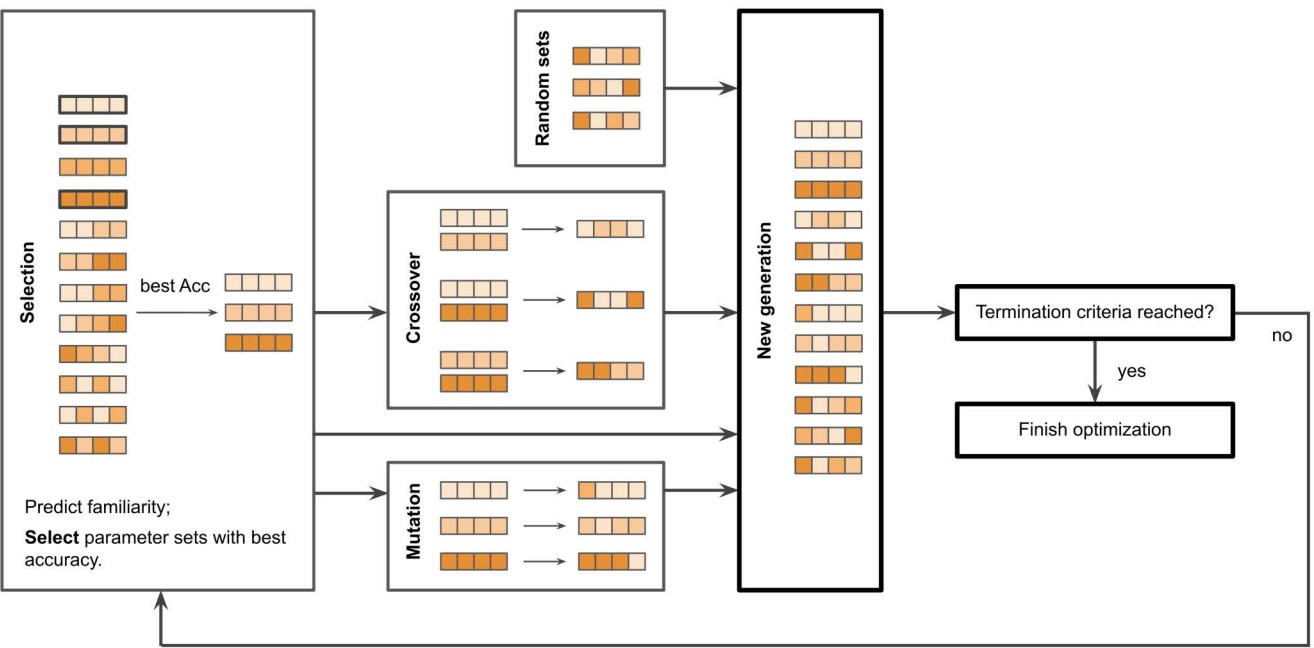

**Fig 7. The genetic algorithm.**

6) Next generation of 12 parameter sets is formed from: 3 best-performing parameter sets of the current generation, 3 sets formed as a result of mutation, 3 sets formed as a result of crossover, and 3 randomly generated sets.

The algorithm described runs either 300 generations in a row, or until the accuracy of 1.0 is achieved by one of the parameter sets, or until in 15 generations in a row the best-performing set of the current generation performs worse or the same as the best-performing set of the previous generation. The classification threshold is determined at each generation iteratively, as the threshold which leads to most accurate classification. When the optimization is finished, the corresponding threshold is fixed for the subsequent experimental simulations.

Importantly, the optimization is performed for a dataset with a fixed repeat interval $R$. Later, in the experimental simulations model performance is evaluated for other $R$ values, which shows the generalization ability of the model.

We also optimized the baseline LSTM model via Adam algorithm, which is a common gradient-based optimization method [49]. The baseline LSTM did not have any Hebbian-like mechanisms; its weights were learned during training and then fixed for the test experiments. During training, a new dataset was generated for each backpropagation step, and no separate validation set was required. LSTM was trained until it reached an accuracy of 97%.

## Izhikevich spiking model

Our network consists of 100 spiking neurons, each receiving independent input from one dimension of the input vector. Spiking dynamics was simulated with an Izhikevich neuron model [31]. The Izhikevich model can approximate the dynamics of different types of neurons as well as a classical Hodgkin-Huxley model [50], but is more efficient, since it has only two differential equations to solve. These equations describe the dynamics of two variables which characterize a neuron state: a membrane potential (voltage) $v_i$ and the recovery variable $u_i$.

$$\dot{v_i} = 0.04v_i^2 + 5v_i + 140 - u_i - I_i + \varepsilon \tag{8}$$

$$\dot{u_i} = a\left(bv_i - u_i\right) \tag{9}$$

$$v_i \leftarrow c \tag{10}$$

$$u_i \leftarrow u_i + d \tag{11}$$

The equations above define how voltage $v_i$ and recovery $u_i$ of neuron $i$ evolve over time. They update every time step and are dependent on the neuron total input $I_i$ and voltage noise $\varepsilon$. The parameter $a$ is a timescale of recovery. In our model, $a = 0.02$. The parameter $b$ describes how sensitive the recovery variable is to the fluctuation of the membrane potential. In our model, $b = 0.2$.

When the voltage reaches the activation threshold 30 mV, we interpret it as a spike event. Note that this threshold rather represents a peak voltage during a spike event. The afterspike neuron dynamics is as follows: the voltage variable is reset according to Eq 8, and the recovery variable gets updated following Eq 9. Noise in the model is described in Eq 12. It is generated randomly at every time step of the dynamical model.

$$\varepsilon = (r - 0.5)\, S_\varepsilon \tag{12}$$

Here, $r$ is a number between 0 and 1 from a uniform distribution. It is adjusted by a voltage noise scaling factor $S_\varepsilon = 0.6$. See S2 Table for a full list of fixed parameters regulating neuronal dynamics.

The total input to the model at every time step comes from two sources: the external stimulus input and lateral input, i.e., input from the other neurons in the model (Eq 13).

$$I = I_{ext} + I_{lat} \tag{13}$$

$$I_{ext} = \delta\,(t - t_i)\,S_{ext} \tag{14}$$

$$I_{lat} = \sum_{j \in J} w_{ij} \tag{15}$$

Izhikevich neurons in the model form one-to-one connections with the binary input vector (see Fig 1), in order to precisely control the input to neurons and make sure that the activity patterns reflect the recurrent dynamics arising from recurrent connections (see Discussion). All feedforward input weights in the model are similar and equal to a constant external input scaling factor $S_{ext}$. Thus, each neuron connected to a non-zero element of the input vector, receives an external input $I_{ext}$ equal to a scaling factor $S_{ext}$ at 100 events evenly distributed per 1000 ms via the Poisson point process.

In Eq 15, recurrent (lateral) input $I_{lat}$ to a neuron i is represented by a summation of weights from the neurons of subset $J$, which are laterally connected to neuron $i$ and emitted a spike at the previous time step. The weights are constantly changing through the ongoing STDP or anti-STDP process.

## Correlated inputs

To generate correlated input patterns, we simplified the procedure from Bogacz & Brown [51], so that it could straightforwardly work with patterns of various sparseness. A binary template pattern was first initialized with a predefined number of active bits, representing a target sparseness level. Each dataset pattern was generated by selecting a subset of random active bits to match those in the template, with the remaining active bits randomly chosen from the remaining non-pattern units. The fraction of active bits inherited from the template, i.e., the overlap between the pattern and the template (Eq 16), was controlled by a parameter $p_{match}$, analogous to the bias parameter in the original procedure [49]. Our adaptation maintains constant sparseness by fixing the number of active bits per pattern. We also report the average pairwise Jaccard similarity – the ratio of shared active bits to the union of active bits – between all pairs of generated patterns (see Eq. 17).

$$\mathbb{E}\left[\text{Overlap}(x, \text{template})\right] = K \cdot p_{\text{match}} \tag{16}$$

$$\text{Jaccard}(A, B) = \frac{|A \cap B|}{|A \cup B|} \tag{17}$$

In Eq. 16, $K$ is the number of active units in each pattern, i.e., the pattern size. In Eq. 17, $A$ and $B$ are the example patterns.

We tested familiarity detection performance under $p_{match}$ values 0.2, 0.4, 0.6, 0.8. See Table 3 for average Jaccard similarity between patterns in the dataset – the ratio of shared active bits to the total number of active bits between every two patterns – generated with different $p_{match}$ values.

## Measuring model performance

The performance of models in the experiments was measured as prediction accuracy balanced by a proportion of novel stimuli in the dataset. Accuracy in the model depends on the true positive and false positive rate of its predictions (Eqs 18–19), following [22].

**Table 3. Jaccard similarity.** Template similarity ($p_{match}$) and corresponding average pairwise Jaccard similarity computed from the generated dataset.

| Template similarity $p_{match}$ | Avg Jaccard similarity (mean±sd) |
|---|---|
| 0.2 | 0.11±0.05 |
| 0.4 | 0.14±0.05 |
| 0.6 | 0.25±0.06 |
| 0.8 | 0.48±0.06 |

Note that additional analyses (connectivity, parameter distribution, performance regression) were only conducted for uncorrelated inputs.

$$f_{new} = \frac{1}{1 + p_{new}} \tag{18}$$

$$Acc = (1 - f_{new}) P_{TP} + f_{new} (1 - P_{FP}) \tag{19}$$

Here, $f_{new}$ refers to the fraction of novel stimuli in the dataset and depends on $p_{new}$ − the probability to generate a new random stimulus while building a dataset. The accuracy of the model predictions $Acc$ takes into account $p_{TP}$ and $p_{FP}$: the probability of correctly classifying a repeated stimulus as familiar, and incorrectly classifying a randomly generated new stimulus as familiar, respectively.

## Regression analysis

To estimate how well different connectivity features influence the LESH generalizability, we conducted a regression analysis using two methods: Linear regression and Decision Tree regression. Both methods were implemented with the use of the *scikit-learn* package in Python [52]. Linear regression was used to model the linear relationship, i.e., a weighted sum of the features, between the connectivity features as predictors and the model generalizability on a task of continual familiarity across the range of repeat interval values as a target variable. In contrast, Decision Tree regression splits the predictor feature space into a series of decision-based hierarchical partitions, in order to reduce variance in the target variable within each partition.

We performed 10-fold cross-validation to ensure the robustness of the analysis and prevent overfitting, on a dataset consisting of 360 samples (the data for all optimization repeat intervals and sparseness levels was combined: 3 repeat interval values x 3 sparseness levels x 40 trials). We evaluated two models: with the target generalizability variable estimated from $R_{sync}$ -based performance, and from rate-based performance.

A set of restrictions was imposed on every model. For Linear regression, we applied the Elastic Net algorithm, which balances L1 and L2 regularization to mitigate overfitting [53]. The Decision Tree models were constrained to a maximum depth of 5 to maintain interpretability and avoid overfitting. Additionally, the grid search was performed on sets of all connectivity features (Gini index, modularity, transitivity, participation coefficient, betweenness centrality) for both Decision Tree and Linear Regression models, to exclude the features which did not contribute to the model performance, thus selecting the connectivity features important for predicting generalizability. For Linear regression, only betweenness-centrality yielded non-zero regression coefficients, whereas for Decision Tree regression the important features were betweenness-centrality, Gini index and transitivity.

Feature importance was also assessed with automated tools provided by *scikit-learn* [52]. For Linear regression, the feature importance was computed as a magnitude of the regression coefficients. For Decision Tree regression, it was calculated as a reduction in a node/split impurity, averaged across all nodes in the tree and weighted by the number of samples per node (Eqs 20–21).

$$FI(i) = \sum_{t \in T} \frac{N_t}{N} \Delta I_t(i)$$

(20)

$$\Delta I_t(i) = I(t) - \left( \frac{N_{left}}{N_t} I(\text{left}) + \frac{N_{right}}{N_t} I(\text{right}) \right)$$

(21)

Here, $N$ stands for the total number of samples in the dataset, $T$ represents a set of nodes in a Decision Tree, $N_t$ is a number of samples per node, and $\Delta I_t(i)$ is a reduction in impurity at the node $t$ due to the split based on the value of feature $i$. Eq 20 defines how the reduction in impurity is calculated: $I(t)$ stands for the impurity of the parent node $t$, $I(left)$ and $I(right)$ represent the impurity of left and right node children respectively. $N(left)$ and $N(right)$ stand for the number of samples in left and right children nodes. The node impurity is calculated as a mean-squared error (MSE). It captures how the MSE decreases due to the specific split.

The performance of regression models was evaluated with the root mean square error (RMSE) and the coefficient of determination ($R^2$). RMSE is a measure of a predictive error which captures the standard deviation of residuals in a regression, and $R^2$ is a goodness-of-fit metric, which quantifies how much variance of the target variable is explained by the model. Thus, a superior model has lower RMSE and higher $R^2$ scores (Eqs 22–23).

$$\text{RMSE} = \sqrt{\frac{1}{n} \sum_{i=1}^{n} (y_i - \hat{y}_i)^2}$$

(22)

$$R^2 = 1 - \frac{\sum_{i=1}^{n} (y_i - \hat{y}_i)^2}{\sum_{i=1}^{n} (y_i - \bar{y})^2}$$

(23)

In both formulas, the differences between true and predicted values of $y$ are computed, and $n$ stands for the number of data points.

## Supporting information

**S1 Table. Fixed model parameters.** Values of model and connectivity parameters fixed in all experiments.
(DOCX)

**S1 Fig. Plasticity regimes do not depend on repeat intervals.** Parameter distributions more strongly differ across repeat intervals than familiarity detection measures ($R_{sync}$ and spike count). Each distribution, represented by a colored curve, includes 20 parameter values received during 20 independent optimization procedures. Color stands for repeat interval used during optimization: orange 3, green 6, blue 10. Colored numbers stand for p-values for differences between measures within every sparseness level, computed via a permutation test with 10000 permutations and Bonferroni correction for multiple comparisons. Data for plasticity types: **A**. Hebbian. **B**. Anti-Hebbian. No differences between $R_{sync}$ and spike count-optimal parameters are significant ($p < 0.05$).
(TIF)

**S2 Fig. Plasticity regimes weakly depend on input similarity.** Parameter distributions differ slightly across input template similarity values. Each distribution, represented by a colored curve, includes 10 parameter values received during 10 independent optimization procedures. Color stands for input correlation values: orange 0.0, green 0.2, blue 0.4, red

0.6, purple 0.8. Colored numbers stand for p-values for differences between measures within every sparseness level, computed via a permutation test with 10000 permutations and Bonferroni correction for multiple comparisons. Data for plasticity types: **A**. Hebbian: values of trace memory are significantly larger for more correlated inputs ( $p < 0.05$ ). **B**. Anti-Hebbian. No differences between $R_{sync}$ and spike count-optimal parameters.
(TIF)

**S2 Table. Fixed Izhikevich parameters.** Values of neuron dynamics parameters fixed in all experiments.
(DOCX)

## Author contributions

**Conceptualization:** Viktoria Zemliak, Pascal Nieters.

**Data curation:** Viktoria Zemliak.

**Formal analysis:** Viktoria Zemliak.

**Funding acquisition:** Gordon Pipa.

**Investigation:** Viktoria Zemliak.

**Methodology:** Viktoria Zemliak, Pascal Nieters.

**Resources:** Gordon Pipa, Pascal Nieters.

**Software:** Viktoria Zemliak.

**Supervision:** Gordon Pipa, Pascal Nieters.

**Validation:** Viktoria Zemliak, Pascal Nieters.

**Visualization:** Viktoria Zemliak.

**Writing – original draft:** Viktoria Zemliak, Pascal Nieters.

**Writing – review & editing:** Viktoria Zemliak, Gordon Pipa, Pascal Nieters.

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
