## [Decision Letter · Decision Letter 0]

30 Mar 2025

PCOMPBIOL-D-25-00093

Continual familiarity decoding from recurrent connections in spiking networks

PLOS Computational Biology

Dear Dr. Zemliak,

Thank you for submitting your manuscript to PLOS Computational Biology. After careful consideration, we feel that it has merit but does not fully meet PLOS Computational Biology's publication criteria as it currently stands. Therefore, we invite you to submit a revised version of the manuscript that addresses the points raised during the review process.

Please submit your revised manuscript within 60 days May 30 2025 11:59PM. If you will need more time than this to complete your revisions, please reply to this message or contact the journal office at ploscompbiol@plos.org. Please include the following items when submitting your revised manuscript:

We look forward to receiving your revised manuscript.

Kind regards,

Rafal Bogacz

Guest Editor

PLOS Computational Biology

Joseph Ayers

Section Editor

PLOS Computational Biology

**Additional Editor Comments :**

The manuscript has been evaluated by 2 expert reviewers who appreciated the clarity of the work, and also made several useful suggestions on how the paper can be further imporved. I would particularly like to encourage the Authors to follow the suggestion of Reviewer 1, and investigate the ability of the model to discriminate familiarity for correlated patterns, because the correlation is present between activities of neurons providing visual input, and many previous models do not work for such correlated patterns. Therefore an ability of a model to discimante the familiarity of correlated patterns is strongly indicates if the model is likely to describe the principles underlying familiarity discrimination in the brain.

**Journal Requirements:**

At this stage, the following Authors/Authors require contributions: Viktoria Zemliak, Gordon Pipa, and Pascal Nieters. Please ensure that the full contributions of each author are acknowledged in the "Add/Edit/Remove Authors" section of our submission form.

4) Please ensure that all Figure files have corresponding citations and legends within the manuscript. Currently, Figures 6, and 7 in your submission file inventory do not have in-text citations. If the figure is no longer to be included as part of the submission, please remove it from the file inventory.

5) We have noticed that you have uploaded Supporting Information files, but you have not included a list of legends. Please add a full list of legends for your Supporting Information files after the references list.

3) If any authors received a salary from any of your funders, please state which authors and which funders.

7) Please ensure that the funders and grant numbers match between the Financial Disclosure field and the Funding Information tab in your submission form. Note that the funders must be provided in the same order in both places as well. Currently, "the German Research Foundation (DFG) − 456666331" is missing from the Funding Information tab.

**Reviewers' comments:**

Reviewer's Responses to Questions

**Comments to the Authors:**

**Please note that the reviews are uploaded as attachments.**

Reviewer #1: Please see attached for my reviews.

Reviewer #2: In attachement

**Have the authors made all data and (if applicable) computational code underlying the findings in their manuscript fully available?**

Reviewer #1: Yes

Reviewer #2: Yes

PLOS authors have the option to publish the peer review history of their article (what does this mean? ). If published, this will include your full peer review and any attached files.

**Do you want your identity to be public for this peer review?** For information about this choice, including consent withdrawal, please see our Privacy Policy .

Reviewer #1: No

Reviewer #2: **Yes: ** Jacques Sougné

**Figure resubmission:**
---

## [Decision Letter · Decision Letter 1]

3 Jul 2025

PCOMPBIOL-D-25-00093R1

Continual familiarity decoding from recurrent connections in spiking networks

PLOS Computational Biology

Dear Dr. Zemliak,

Thank you for submitting your manuscript to PLOS Computational Biology. After careful consideration, we feel that it has merit but does not fully meet PLOS Computational Biology's publication criteria as it currently stands. Therefore, we invite you to submit a revised version of the manuscript that addresses the points raised during the review process.

Please submit your revised manuscript within 30 days Sep 02 2025 11:59PM. If you will need more time than this to complete your revisions, please reply to this message or contact the journal office at ploscompbiol@plos.org. Please include the following items when submitting your revised manuscript:

We look forward to receiving your revised manuscript.

Kind regards,

Rafal Bogacz

Guest Editor

PLOS Computational Biology

Hugues Berry

Section Editor

PLOS Computational Biology

**Additional Editor Comments :**

Thank you for updating the manuscript so thoroughly according to Reviewers comments. Reviewer 2 checked the revisions, and you can see their comments below. Reviewer 1 was not available, so I checked the manuscript myself, and have a few comments listed below.

Minor:

l.89-90: “Li and Bogacz”-> “Li et al”. Please also note that a more recent version of reference 27 is available as an arxiv preprint – please update the reference to this newer version.

Figure 1C: This cartoon does not illustrate the correlation well – For the first panel, the correlation is -1 rather than 0. Please update the panels so the values of correlation match the cartoons.

l. 281: “HebbFF on the data of sparseness 0.0” – but definition of sparseness in line 188 suggests that 0.0 correspond to all pattern being identical, so the familiarity discrimination does not make sense. Please clarify.

lines 356 and 390 and 584: “Anti-Hebbian models” – Please make it more specific, e.g. “recurrent networks with anti-Hebbian plasticity”, to avoid confusion with the anti-Hebbian model (Bogacz et al. 2003), which have different properties.

Typos:

line 126: Mathematical symbol not properly displayed.

l.267-273: “HebFF”, “HebbFF”, “HEbbFF” – please use consistent spelling of the name of this model.

l.863: “Bogacz. &” -> “Bogacz &”

**Journal Requirements:**

1) We have noticed that you have uploaded Supporting Information files, but you have not included a list of legends. Please add a full list of legends for your Supporting Information files after the references list.

**Reviewers' comments:**

Reviewer's Responses to Questions

Reviewer #2: The authors have done an impressive job both in extending their research and improving the readability. They really took into account reviewer comments, thanks.

I have no major comment to make, only a few minor remarks.

L126: the tau is in an invisible character

Figure 2 C The colors of LSTM and HebbFF curves are too similar to be discriminated. I had to read the text carefully to recognize which was which.

L526: in the paragraph “In Hebbian networks…” trace memory not race memory.

Page 27, I wonder if it would not be better to insert supplementary material inside the main text. The first time I read the last paragraph (L579-587), I didn't understand it until I dug into the supplementary information.

L751, first paragraph (see Table 2) and not (se Table 2).

**Have the authors made all data and (if applicable) computational code underlying the findings in their manuscript fully available?**

Reviewer #2: Yes

PLOS authors have the option to publish the peer review history of their article (what does this mean? ). If published, this will include your full peer review and any attached files.

**Do you want your identity to be public for this peer review?** For information about this choice, including consent withdrawal, please see our Privacy Policy .

Reviewer #2: No

**Figure resubmission:**
---

## [Editor Report · Decision Letter 2]

8 Jul 2025

Dear Ms Zemliak,

We are pleased to inform you that your manuscript 'Continual familiarity decoding from recurrent connections in spiking networks' has been provisionally accepted for publication in PLOS Computational Biology.

Best regards,

Rafal Bogacz

Guest Editor

PLOS Computational Biology

Hugues Berry

Section Editor

PLOS Computational Biology

Thank you for adapting the manuscript according to the comments.

In Figure 1C the values of correlation still do not correspond to the patterns, for example, the patterns labelled as 0.2 and 0.4 have actually negative correlation. When you submit your final version, please correct the patterns in this figure, so they match the correlation labels above them

---

## [Editor Report · Acceptance letter]

PCOMPBIOL-D-25-00093R2

Continual familiarity decoding from recurrent connections in spiking networks

Dear Dr Zemliak,

I am pleased to inform you that your manuscript has been formally accepted for publication in PLOS Computational Biology. Your manuscript is now with our production department and you will be notified of the publication date in due course.

With kind regards,

Zsofia Freund
